# An Exploratory Study into the Influence of Sex on Body Measurements, Carcass Weights and Meat Yields of Giraffe (*Giraffa camelopardalis angolensis*)

**DOI:** 10.3390/foods10102245

**Published:** 2021-09-22

**Authors:** Bianca L. Silberbauer, Phillip E. Strydom, Louwrens C. Hoffman

**Affiliations:** 1Department of Animal Sciences, University of Stellenbosch, Private Bag X1, Matieland, Stellenbosch 7602, South Africa; Biancasilberbauer95@gmail.com (B.L.S.); pestrydom@sun.ac.za (P.E.S.); 2Center for Nutrition and Food Sciences, Queensland Alliance for Agriculture and Food Innovation (QAAFI), The University of Queensland, Digital Agricultural Building, 8115, Office 110, Gatton 4343, Australia

**Keywords:** body measurements, game meat, giraffe, yield

## Abstract

Various body measurements and commercial carcass yields of relatively young (2½–6 yrs old) giraffe (*Giraffa camelopardalis angolensis*) were investigated to quantify the effect of sex there upon. Eight male and eight female giraffe were culled by standard practice in Namibia, where body and horn measurements were taken, before the carcasses were dressed. There were no significant differences between the mean dead weights of the two sexes (bulls = 691.1 kg; cows = 636.5 kg; *p* = 0.096), the only body measurements found to differ significantly were those of the forelegs, with the shoulder to hoof (*p* = 0.046) and the knee to hoof (*p* = 0.025) both being significantly longer in the bulls. The horn measurements were all found to be significantly larger in the bulls than the cows even at this young age. The neck weight as a percentage of the carcass weight was found to be significantly heavier for the bulls compared to the cows, however, the back percentage values were significantly heavier in the cows than the bulls. There was a strong positive correlation between the body weight and most of the body lengths, as well as between most of the individual body measurements. The giraffe used had an average age of 3.7 years old, and had therefore not yet reached their growth plateau, which may be why sex had no influence on most of the body measurements recorded.

## 1. Introduction

Giraffe are the tallest land mammal walking our planet [1,2], standing at up to nearly 6 m tall, yet relatively little is known about these African giants, especially in the wild. Although many South African and Namibian farmers keep them, not many actively farm them in a structured breeding program as they may do with game species which are farmed for hunting and meat. However, there is often the necessity to cull some of the population as in farm situations, they will not have any predators, and their population numbers must be controlled in order to prevent surpassing the carrying capacity of the land. Although game farms usually cover large areas, particularly in Namibia, game are still fenced in and can therefore not migrate across the land when food become scarce. In addition, due to the huge variation in species (size and species), and because of their dietary preferences (browsers vs. grazers and selectivity), it is of paramount importance to manage animal numbers in order to ensure the animals’ well-being and manage the ecological balance [3]. Relocating game from overstocked farms is expensive, risky and has limited scope because demand cannot keep up with supply (breeding rate); hence the necessity to selectively cull animals as required by good management practices [3]. In Namibia, landowners, or custodians of land with fenced and non-fenced (open) land can apply to the authorities for a “shoot-and-sell permit” (day harvesting) or a night-culling permit to harvest game for commercial meat production. For own consumption of huntable game, no permit is required—only if the meat is to be transported off the property for commercial purposes. As a species, giraffe is listed as Vulnerable on the IUCN Red List of Threatened Species, although the Angolan giraffe (*G. camelopardalis angolensis*) found in Namibia is the second most abundant subspecies after the after the South African giraffe (*C. c. giraffa*). This status as well as the numbers of giraffe on a specific property is taken into consideration when the Namibian authorities evaluate the local population before issuing a “shoot-and-sell permit” (personal communication, owner of Mount Etjo Game Farm). On the farm on which the giraffe for this trial were harvested their total population has increased from approximately 900 to 1100 giraffe between 2013 and 2017, despite culling between 50 and 65 predominantly young bulls each year, and about 75 in 2018. This results in a large quantity of meat for which there is little to no market, since there is very little known about its quality as yet; Hall-Martin [4] reported on a basic study on the carcass composition as well as muscle fibre diameter of giraffe, however, there has not been any follow up study on their meat quality since then. Anecdotally, it seems as if giraffe meat is generally only used in processed products such as boerewors/sausage when sold commercially. On the farm where the giraffe were harvested, tourists were willing to consume giraffe meat and saw it as part of the “Africa” experience (personal communication, owner of Mount Etjo Game Farm); this is in agreement with an earlier study amongst tourists visiting South Africa and their perceptions of consuming game meat [5] as well as that of South African citizens [6], although in the latter study there were differences in perception depending on cultural/ethnic differences. In the former study [5] among sixty tourists from Europe, respondents indicated game meat as the meat type they most favoured to order in restaurants in South Africa. They were aware of the health benefits related to game meat consumption and the culling of game animals did not concern most of them as long as it was performed in an ethical way. Local consumers across various ethnic groups [6] were mostly indifferent or positive about the concept of culling of game animals for the purpose of protecting both the habitat and the diversity of game species in Africa.

The tremendously long neck of the giraffe has been the muse of biologists for centuries, but every great muse has its mystery, and there is still no definitive answer to why the giraffe has such a long neck. It has always been the general assumption that the extreme height of the giraffe is to enable it to reach browse beyond the reaches of other browsers that they may compete with, as with other species where a long neck has evolved in order to access more food [7]. However, this theory has been cast under shadow by du Toit [8] who noted that giraffe seldom browse with the full reach of their neck, but rather tend to browse at the same level as many other browsers do, as Young and Isbell [9] and Leuthold and Leuthold [10] confirm with their findings which show that giraffe prefer to browse at their shoulder height for most of the time.

The other hypothesis for why giraffe developed such a long neck was that bulls with longer necks had a sexual advantage as they use their necks in fighting for females [11], however, a study on a population of Zimbabwean giraffe (*Giraffa camelopardalis giraffe*; [12]) found evidence of limited sexual dimorphisms of neck and leg length, and head and neck mass, and those that were found could be explained as generic male-female differences, as occur in most species. However, evidence to the contrary was noted in a Namibian giraffe population (*G. c. angolensis*), where it was found that neck size increased for bulls throughout their lives, whilst it plateaued for cows [11]. 

A large proportion of research that does exist on giraffe was done on captive giraffe kept in zoo environments, and there is substantial evidence that giraffe behaviour, growth, and general performance, from feeding and drinking patterns, to longevity and onset of puberty, is markedly different between captive giraffe and giraffe in the wild [1,4,13]. This, therefore, must be considered when evaluating any of the findings where only captive giraffe have been used.

In 1977, two studies were conducted on a group of giraffes [4,14]. One on body measurements and leg weights as a prediction of the total weight of the giraffe, in order to determine the size of a giraffe if only portions are left of the carcass after being killed by predators. The other was on meat production, quality, and muscle fibre type of giraffe. Both studies were limited, however, by the lack of proper facilities, forcing them to be performed in the field, and with a scale unable to weigh the whole dead giraffe. This study aims to clarify some of these earlier findings [4,14] and broaden the knowledge base on these African giants as well as provide some base line information on their potential meat yields.

## 2. Materials and Methods

The Sixteen giraffe (eight bulls, eight cows) were obtained from Mount Etjo Game Farm in July of 2018 in the Otjozondjupa region of Namibia; these animals comprised part of an annual cull carried out on the farm. All animals were aged by professional hunters and all but one were estimated to be between two and a half years and six years old (using body weight as determinant from results of Hall-Martin and colleagues [4,14]—data not shown), G13, however, was judged to be a mature female approximately nine years old, and was consequently removed from the analyses, in order to prevent age from skewing the data. The giraffe were culled by a head shot and then bled (Ethical approval: ACU-2018-7366, Stellenbosch University; Namibian Shoot and Sell Permit number: 118690) before being measured as described below. The giraffe were then weighed to give a dead weight (live weight less blood loss), before being transported to the abattoir where the carcasses were dressed as described by Ledger [15] and cut into sections similarly to Hall-Martin [4,14] for cooling in the cold-room. 

The measurements were taken with a soft measuring tape with the giraffe lying flat on its side with neck and legs extended, except where otherwise stated, as follows (Figure 1): 

Body length (BDL) measured from the point of the sternum protruding furthest from the chest, over the shoulder and side, to the dorsal point of the hip, the pin.

Back length (BL) was measured from the top of the withers (the highest point of the spine at the third thoracic vertebra), along the curve of the back to the base of the tail.

Body depth (BD) was measured from the top of the withers around the girth to the mesodistal point of the sternum directly below the withers.

The girth (G) measurements were taken when the giraffe was hanging by the neck, and were taken around the girth line just behind the withers and just behind the forelegs and around the mesodistal point of the sternum.

Neck circumference (NC) was taken around the base of the neck at the broadest part of the neck where it meets the shoulders.

Neck length (NL) was taken from the base of the skull along the neck to the top point of the withers.

The shoulder to hoof (S-H) measurement was taken in a straight line from the top of the withers to the bottom of the hoof when extended as though flat on the floor as when the giraffe is standing.

Knee to hoof (K-H) measurement was taken from the mid-point of the knee to the bottom of the extended hoof in a straight line.

Pin to hoof (P-H) measurements were taken in a straight line from the top point of the hip, the pin, to the bottom of the extended hoof.

Hock to hoof (H-H) measurements were taken from the point of the hock to the bottom of the extended hoof in a straight line.

The scrotal circumference on bull animals was taken around the widest section of the scrotum.

The horn length was taken in a straight line from the mid-point at the base of the horns where the two meet to the tip of one of the horns and assumed to be the same for both. 

Minimum horn circumference was taken around the narrowest point of the horn. 

The maximum horn circumference was taken around the broadest part of the horn at the base where it joins the skull.

The tip to tip measurement was taken from the top most point of one horn to the same point on the other horn in a straight line.

On arrival at the abattoir the pre-rigor (within two hours post-mortem, the carcasses were deemed to be pre-rigor as the legs and neck could be moved freely) giraffes were hoisted by a crane with a hook placed through the hock of one hind leg, they were then skinned and eviscerated as described by Ledger [15]. The carcass was then split into eight sections, the two forelegs, two hind legs, the neck, two rib racks and the spinal column including the tail. These sections were cut as described by Hall-Martin [3] except for the cut between the ribs and the spinal column. 

The forelegs were removed by cutting from the olecranon process of the ulna along the *tensor fasciae antibrachii* muscle to the caudal angle of the scapula, thus cutting the *latissimus dorsi* muscle where it runs inferior to the caudal edge of the *triceps* muscle. The pectoral muscles were then severed close to where they join the foreleg as it was being held out from the body. The cut was then continued along the cranial edge of the *biceps branchii* muscle and the *supraspinatus* muscle around the dorsal end of the scapular cartilage, through the *trapezium* muscle, before cutting through the last of the connective tissue connecting the cartilage of the scapular to the thorax. 

The hind legs were removed by making a cut along the lateral edge of the sacrum, severing muscle attachments with a mesodistal cut between the *tuber coxae* and the *tuber ischia*, removing the muscles cleanly from the bone and obturator membrane. The muscle attachments of the *tensor fasciae* muscle and the patellar ligament were then severed with a ventral cut along the caudal edge of the *tuber coxae*. The head of the femur could then be disarticulated and the remaining connective tissue severed.

The head was removed at the axis-atlas joint from the neck. The neck was then removed from the thorax by cutting between the seventh cervical vertebra and the first thoracic vertebra. 

A cut was made through the middle of the abdominal muscles from just before the pelvis through the heads of the thoracic ribs and down through the sternum, splitting it ventrally. The organs were then removed from the thoracic cavity. The cut was extended towards the spine from the pelvis to where it met the edge of the *longissimus lumborum* muscle, a fine-toothed chainsaw was then used to cut through all the ribs along the outer edge of the *longissimus lumborum* muscle and *longissimus thoracis* muscle. 

This left the spinal column, from the first thoracic vertebra to the end of the tail, with the *longissimus lumborum* muscle, *longissimus thoracis* muscle and *psoas major* and *minor* muscles (fillet) still attached (called the “back”), unlike in Hall-Martin’s procedure [4,14].

The ossicones were removed from the rest of the skull with a saw, they were then frozen, and a micro-computerized tomography (CT) scan was performed on those of the largest bull and cow, which was G13, the mature female, respectively. The ossicones were kept frozen and secured in a manner that the scan would revolve around the centre line of the ossicone itself.

Each carcass section was weighed as the warm carcass weight, before being placed into a fridge for approximately 24 h, after which they were weighed again (post rigor) for a cold carcass weight.

Statistica was used to perform analyses of variance on all the variables measured by means of the General Linear Model (GLM). The differences between sexes was tested for by means of the null hypothesis H_0_: µ = µ_0_ and the alternative hypothesis H_a_: µ ≠ µ_0_. This was done by performing contrast analyses and estimated least squares means (± standard error (SE)) as reported in the tables. For all analyses, age was included as a covariate, meaning that all the LS means mentioned above, were adjusted for age. The variables were accepted to be significantly different if the probability of rejection of H_0_ was less than 5% (*p* < 0.05) for sex. Pearson’s correlation coefficients were also calculated between live weights and the various body measurements, as well as between the body measurements.

## 3. Results

As mentioned, one cow was estimated to be significantly older (9 years) than the rest of the cows and her data was subsequently removed from the data analyses.

The only significant difference between the body measurements of the two sexes were those of the forelegs, with both the shoulder to hoof (*p* = 0.046) and the knee to hoof (*p* = 0.025) being significantly longer in the bulls than the cows (Table 1; Appendix A), as well as all of the horn measurements which were also significantly larger in the bulls (*p* ≤ 0.05). 

There was a moderate to strong positive correlation between dead weight and all body measurements, except girth, which had only a moderate positive correlation (r = 0.438; *p* = 0.10) (Table 2). Age had a moderate to strong positive correlation with all body measurements, other than the girth (r = 0.353; *p* = 0.20). In general, the correlations between girth measurements and other body measurements were not as strong as correlations between other measurements including body depth; the latter was expected to be high as they were taken from similar places. 

There was a strong positive correlation between all body measurements (r > 0.600), except for the girth and the scrotal circumference correlations with back, leg and neck lengths. The scrotal circumference was however, strongly correlated with body length and depth, neck circumference, and interestingly horn length. 

The horn measurements were not very strongly correlated with the body measurements but did all have a moderate to strong positive correlation with the foreleg measurements. 

The bulls were not found to be significantly heavier than the cows (*p* = 0.096) (Table 3), with a dead weight of 691.1 ± 45.5 kg (min = 562.3, max = 927.1) whilst cows had a mean dead weight of 636.5 ± 33.8 kg (min = 508.4, max = 747.5). Despite this difference not being statistically significant, it does seem as though the males tend to be heavier; this may be as they are pubescent animals, at the inflection point of their growth curve and they are just beginning to show sexual dimorphisms. The dressed carcass weights, however, did tend towards a significant difference between the sexes, which was more pronounced in the cold carcass weights (*p* = 0.053) than the warm carcass weights (*p* = 0.063) (Table 3). The dressing percentages did not differ significantly between the two sexes though (*p* = 0.982), with dressing percentages of 56.7% and 56.8% of bulls and cows, respectively (cattle dressing percentages generally accepted to average between 58–62%). The only significant differences between the sexes for carcass sections as a percentage of the whole carcass were for the neck (*p* = 0.005) with bulls averaging 44.56 ± 3.6 kg (min = 31.3, max = 63.4) and cows 34.46 ± 1.3 kg (min = 29.6, max = 41.7), and the back where the cows’ backs made up a larger percentage of the carcass than the bulls’ (*p* = 0.026). However, as pertaining to the actual weights, cows’ backs averaged 54.79 ± 2.4 kg (min = 45.0, max = 62.6) and bulls’ backs averaged 54.13 ± 5.0 kg (min = 34.3, max = 81.3)—it is found that they do not differ much.

## 4. Discussion

The method of culling was effective, as every animal was dropped by a head shot, and only one was injured, sustaining a glancing shot to the neck, but the next shot (within 30 s of the first) was a good head shot and it dropped immediately. This meant that no meat had to be discarded due to damage caused by the culling method.

The only significant differences between any of the body measurements (Table 1), between the two sexes, were those of the forelegs. The measurements from shoulder to hoof and knee to hoof were both significantly longer in the males than the females, this seems to be supported by what Mitchell et al. [12] reported, although they did not report age in their study. They did; however, report that for males and females, when increasing from 100 kg to 1100 kg live weight, males had a 2.01-fold increase in foreleg length, while females only had a 1.69-fold increase.

The lack of any other significant sexual dimorphisms in length, including the neck, is supported by what Hall-Martin [4] reports on giraffe of this age. Mitchell [12] found that, contrary to what Simmons and Scheepers [11] reported on males having larger necks for sexual advantages, when correcting for body weight, males and females had no significant differences in neck length, which agrees with the findings of this study, that, despite the lack of sexual dimorphism in length, there was a dimorphism in weight.

Hall-Martin’s study [4] was carried out on a group of Transvaal Lowveld (now Mpumalanga, South Africa) giraffe, of ages ranging from birth to approximately 23 years old. He found that until about five years of age, male and female giraffe show little dimorphisms in height, and that they reach a plateau in growth at about 11 years old in cows and about 12 years old in bulls. However, as his study was only performed on one sub-species of giraffe and a relatively small group of giraffe (*n* = 53, of which 27 were male and 26 female) considering the age spread from birth to 23 years of age, these ages cannot be taken as the rule, especially not across all sub-species of giraffe. However, if using Hall-Martin’s study [4] as a reference for the giraffe used in this trial, these Namibian were predominantly pubescent giraffe around the inflection point of their sigmoidal growth curve, as their average age was approximately 3.7 years. One cow was found to be in an early pregnancy, which supports the assumption that they were not immature giraffe. This validates the findings of sexual dimorphisms, and trends towards dimorphisms, as the giraffe move from puberty towards maturity.

There was a low correlation between the girth measurements and the other body measurements which could be due to errors in these measurements as the girth measurements were the only ones taken when the giraffe was hanging from the neck. It was not possible to get the measuring tape under the body of the giraffe when it was lying on its side, and it should be noted that when they were hung in this manner, the skin would form folds around the shoulders, which would have affected the measurements, as the folds were not uniform for each animal. The contents of the chest cavity may also have shifted downwards thus changing the way the rib cage sits when compared to when the giraffe was lying on its side as when other measurements were taken.

From the CT scans (Figure 2) it could be seen that the bull’s ossicones are more densely ossified than the cow’s. Bulls use their horns to fight and must therefore be able to withstand a huge amount of force as they slam them into their opponent. From Table 1 it can also be seen that the horns are substantially larger in all dimensions, in bulls than cows, with the horn base where the maximum horn circumference was measured as well as the horn length being the most significantly larger. As cows do not use their horns to fight, they have very limited use for them, and thus larger, heavier horns will be a disadvantage for them, as they must carry this weight at the top of their long necks.

The dead weight reported of the giraffe in this study is not a true representation of the live mass of the animal, as it does not account for the blood loss, since they were only weighed after bleeding (Table 3). However, this can be estimated at ~21 kg, as it is reported as being approximately 3–4% of the live mass in large mammals; it was found that blood made up 3.3–3.9% of live mass in dairy cattle [16]. The gut fill of the animal also affects the live mass value, and this depends on the eating and drinking habits of the species, as well as the time of day that the animals were culled relative to this feeding pattern. As giraffe can go for days without drinking [1], it would be very hard to standardise the time from drinking to time of death and thus control water content of the gut fill. The impact of gut fill on live weight is also affected by their feed intake as most of their water requirements are in fact satisfied from the water in the leaves that they eat, therefore the browsing pattern and resultant gut-fill should be taken into account as it has been reported that gut fill can have an influence of up to 20% of live weight in livestock [17]. When comparing dressing percentage with domestic species it must be taken into consideration that domestic species are normally fasted for 12–24 h before slaughter, this will reduce their gut-fill substantially in the domestic species and increase their dressing percentage by up to 4% [18].

Body weight of giraffe also varies with season, due to availability of food however, as all giraffe in this trial were culled over a two-week period in July/August of 2018 (which is late winter in Namibia), this affect should be minimal in this investigation.

The dead weight of the bulls and cows did not differ significantly which appears to be in contradiction with the findings of von La Chevallerie [19], where he reported that for most African ungulates the male is significantly heavier than the cow, which has been supported in a number of studies on giraffe [4,14]. However, the majority of the giraffe used in this study were young (average age of ~3.7 years old); the approximate age at which they reach puberty. According to the study on Transvaal Lowveld giraffe, male and female live weights are still similar until about 4.5 years of age, and only really start to show sexual dimorphisms after this, with cows having a lower growth rate and plateauing at a younger age at about 11 years (~850 kg), and bulls plateauing around 12 years at a weight generally about half a tonne heavier than the cows (~1400 kg) [4]. The difference between the dressed carcass weights of the bulls and cows did tend towards significance, however, this may in part be due to the fact that the female reproductive system may weigh more than that of the male. The dressing percentages did not differ significantly between the sexes.

Only one of the cows was found to be pregnant, which will also impact her dress-out percentage as the foetus and amniotic sac were recorded to weigh 61.15 kg which is 8.18% of her dead weight (Table 3). However, with a dress-out percentage of 52.28% she was in alignment with the other dress-out percentages.

The only significant differences of carcass section weights between the two sexes were for the neck weight, this was expected, and for the back, which was unprecedented. The difference in neck weights could be expected as the bulls use their necks for fighting and consequentially need a strong, well ossified and muscled neck to withstand the force of hitting the neck and head into rival bulls. This is in line with reported findings on the respective weights of bulls’ and cows’ necks [1,11]. Although the respective neck lengths and circumferences (at the base of the neck, it was not measured along the neck) of bulls and cows did not differ significantly, the difference in weight may be explained by the respective densities of the vertebrae of the sexes. It has been found that the vertebrae, especially trending towards the top of the neck of giraffe are less dense than the other bones, which is more pronounced in cows as their necks do not need to withstand the clubbing force of being swung into another giraffe [20]. Therefore, their long necks are as light as possible, minimising the weight that needs to be supported at such a great height. The back on the other hand, was found to make up a greater percentage of the carcass of the cows than the bulls. However, when looking at the actual weights of the backs it was found that there was a very minimal difference between the sexes, therefore, the significance in the percentage is as a result of the necks of the bulls weighing significantly more than cows, and this percentage has to be compensated for in the females. A biological explanation of the higher percentage for the females may be in order to carry a calf; the cows need to have a stronger back than the males.

The necks made up 11.3 ± 0.369% and 9.9 ± 0.332% of the total carcass weight for bulls and cows respectively (Table 3), and of this a large portion is bone, as their vertebrae are greatly enlarged in comparison to other ungulates of similar size, such as eland (*Taurotragus oryx*) or buffalo (*Syncerus caffer*) (L.C. Hoffman personal observation). The little meat that is on the neck is highly sinuous as thick strong tendons and ligaments are necessary to control this ungainly neck and the large head perched on top, therefore despite making up carcass weight it would not add much value to the carcass. The hind legs made up 34.1 ± 0.398% and 34.5 ± 0.306% of the carcass for bulls and cows respectively, and as this contains a large proportion of the prime cuts, this is positive, especially as the meat to bone ratio in the hindquarters is generally favourable. The forelegs made up 26.4 ± 0.224% and 25.8 ± 0.391% of the bull and cow carcass weights respectively, however, since the scapula makes up a fair portion of this weight, there is a less favourable meat to bone ratio here than in the hind legs. The back holding potentially the two most sought after and valuable cuts, the fillet, and the loin, made up 13.8 ± 0.49% and 14.9 ± 0.41% of the carcass in the bulls and cows respecti+vely. The loin, made up of the *longissimus lumborum* muscle and *longissimus thoracis* muscle was found to be riddled with sinew as well, with thick tendons running throughout this cut. This is due to the fact that the tendons supporting the neck originate from all along the back, thus anchoring it for better leverage. It was also observed that the structure of the muscle as a whole was loose; it was not always clear to see the direction of the muscle fibres which seemed to run in many different directions throughout the muscle. The fillet on the other hand was smaller than expected, but not rife with tendons as was the loin. The fillet was also more triangular and looser in shape whereas in other animals it tends to be more tubular and more compact. Similar structures for both the back muscle/loin and fillet have been observed in the elephant’s back by the corresponding author. Another observation was that the back was shorter (~105 cm) than may have been expected (Table 1), when compared to the length of their neck and legs; in livestock species the back tends to be longer giving a longer length of the high value dorsal muscles. The ribs making up 14.4 ± 0.295% and 15.0 ± 0.297% respectively of the total carcass for bulls and cows, as with any ribs, are predominantly made up of bone with very little meat on the sides. 

The carcasses tended to have a very visible white collagenous subcutaneous layer under the skin, which was stretched very tightly over the muscles, as soon as it was punctured the underlying muscle would bulge out of the hole created. Although the giraffe generally had very little fat, the carcasses did have a fairly localised fat covering along the back, which tended to be whiter in colour, as well as a thicker yellowish fat layer around the withers. The kidney and caul fat depots were prominent and were white and hard. There was no visible intramuscular fat observed in the meat. 

## 5. Conclusions

This study was performed on a small group of giraffe, and on predominantly pubescent giraffe around the inflection point of their sigmoidal growth curve, it could be of interest to extend this information with future studies on a wider range of ages to create a broader result and to create a more robust carcass yield and growth curve for giraffe; the value of these regression equations will obviously be determined by the accuracy of age determination, a challenge in free roaming wild giraffe. It may also be of interest to develop regression curves of the various subspecies to determine whether they differ. This study may, however, prove valuable to other farmers that also cull predominantly young bulls, as it gives an indication of the carcass yields. 

Giraffe have a favourable dressing percentage, with the more traditionally valuable cuts making up a large percentage, however, it may be of interest to perform a block test on the whole carcass in order to see actual percentages of clean meat, bone and sinew, so that a better idea of the value of the carcass could be established. On-going research should also evaluate the quality aspects of these cuts so as to give guidance on how they could be marketed. It is also interesting to note that at an average age of ~3.7 years, there is very little difference between the carcasses of the bulls and cows.

## Figures and Tables

**Figure 1 foods-10-02245-f001:**
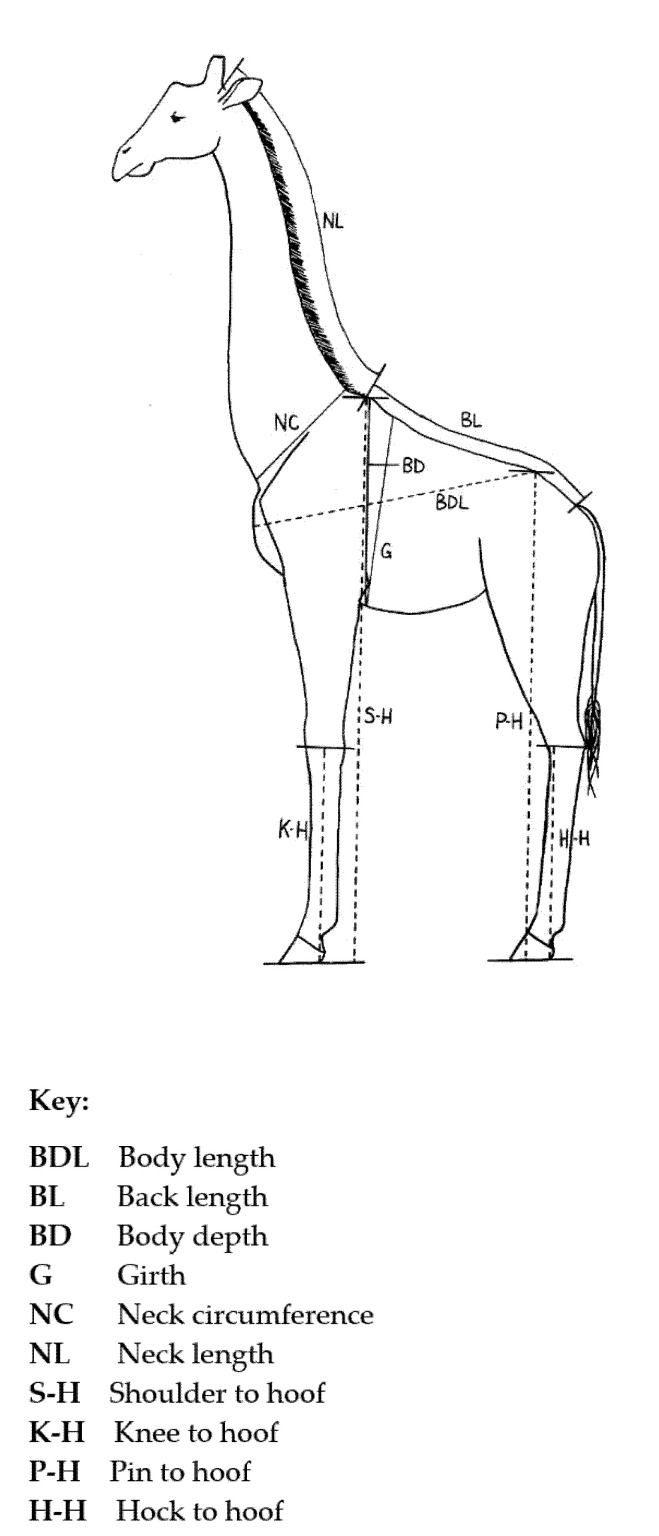
Diagram of the locations where body measurements were taken on the giraffe (*Giraffa camelopardalis angolensis*).

**Figure 2 foods-10-02245-f002:**
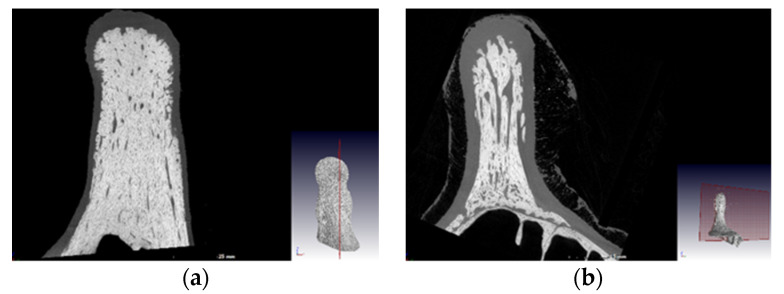
Computerized tomography (CT) scan images of the internal structure of the ossicones of (**a**) a giraffe bull and (**b**) a giraffe cow.

**Table 1 foods-10-02245-t001:** Body and horn measurements of ~3.7 year-old male and female giraffe (*G. c. angolensis*).

Body Measurements	Male (*n* = 8)			Female (*n* = 7)			*p*
	Mean	S.E.	Range	Mean	S.E.	Range	
Dead weight (kg)	691.1	45.465	562–927	636.5	33.764	508–747.5	0.096
Body length (cm)	160.1	3.346	151–177	157.7	4.190	136–171	0.391
Back length (cm)	105.3	4.872	83–123	102.6	3.054	90–114	0.534
Body depth (cm)	123.0	3.295	109–135	122.4	2.759	109–129	0.671
Girth (cm)	236.4	8.508	202–277	230.4	4.551	210–246	0.499
Neck length (cm)	134.5	4.702	119–155	134.6	4.314	115–152	0.913
Neck circumference (cm)	135.4	5.904	117–161	132.0	4.077	119–145	0.453
Shoulder to hoof (cm)	272.9	6.306	256–304	260.1	4.688	239–276	0.046
Knee to hoof (cm)	92.8	1.770	88–101	88.3	1.340	82–92	0.025
Pin to hoof (cm)	233.0	4.702	218–253	229.0	4.309	209–240	0.381
Hock to hoof (cm)	100.5	2.471	94–113	96.4	1.837	88–102	0.137
Scrotal circumference (cm)	26.5	1.041	23–31	-	-	-	-
Horns:							
Length (cm)	16.7	0.756	13.5–19	11.6	0.404	10–13	<0.001
Minimum circumference (cm)	15.8	0.841	13.5–20	11.9	0.322	10.5–13	0.001
Maximum circumference (cm)	32.2	1.069	28–36	22.4	0.948	19–26	<0.001
Tip to tip (cm)	17.1	0.601	15–19.5	13.4	1.152	9.5–16.5	0.005

**Table 2 foods-10-02245-t002:** Pearson Correlation coefficients (r) of the body and measurements of ~3.7-year-old giraffe (*G. c. angolensis*).

	Dead Weight	Age	Body Length	Back Length	Body Depth	Girth	Neck Length	Neck Circumference	Shoulder to Hoof	Pin to Hoof	Knee to Hoof	Hock to Hoof	Horn Length	Min Horn Circumference	Max Horn Circumference	Tip to Tip
Age	0.758															
Body length	0.921	0.724														
Back length	0.768	0.508	0.799													
Body depth	0.876	0.741	0.913	0.766												
Girth	0.438	0.353	0.496	0.496	0.260											
Neck length	0.686	0.490	0.750	0.813	0.699	0.520										
Neck circumference	0.894	0.644	0.820	0.658	0.867	0.390	0.647									
Shoulder to hoof	0.899	0.583	0.852	0.829	0.773	0.567	0.793	0.837								
Pin to hoof	0.872	0.581	0.905	0.844	0.889	0.482	0.825	0.900	0.926							
Knee to hoof	0.818	0.508	0.770	0.727	0.683	0.521	0.763	0.780	0.951	0.864						
Hock to hoof	0.786	0.474	0.767	0.682	0.710	0.443	0.758	0.818	0.891	0.907	0.926					
Horn length	0.490	0.396	0.384	0.228	0.300	0.355	0.178	0.369	0.558	0.376	0.638	0.514				
Min horn circumference	0.451	0.167	0.363	0.469	0.421	0.111	0.279	0.293	0.542	0.430	0.493	0.390	0.673			
Max horn circumference	0.364	0.098	0.285	0.221	0.216	0.243	0.215	0.260	0.489	0.345	0.553	0.443	0.878	0.802		
Tip to tip	0.405	0.374	0.452	0.266	0.388	0.296	0.265	0.346	0.438	0.406	0.569	0.492	0.826	0.493	0.769	
Scrotal circumference	0.822	0.846	0.751	0.271	0.692	0.378	0.346	0.759	0.599	0.534	0.498	0.465	0.677	0.210	0.111	0.265

0.000–0.399, Weak positive correlation; 0.400–0.599, Moderate positive correlation; 0.600–0.799, Strong positive correlation; 0.800–1.000, Very strong positive correlation.

**Table 3 foods-10-02245-t003:** Mean (±SE) carcass yields of ~3.7 years old male and female giraffe (*G. c. angolensis*).

Parameter	Male (*n* = 8)	Range (min–max)	Female (*n* = 7)	Range (min–max)	*p*
Dead weight (kg)	691.1 ± 45.465	562.3–927.1	636.5 ± 33.764	508.4–747.5	0.096
Dressed weight (kg)	400.4 ± 28.622	314.5–543.5	366.4 ± 15.576	295.7–424.3	0.063
Dressout ^a^ (%)	56.7 ± 0.850	51.6–59.2	56.8 ± 1.278	52.3–63.4	0.982
Cold carcass weight (kg)	393.1 ± 28.522	310.1–535.8	359.5 ± 14.490	290.9–407.3	0.053
Moisture loss in chiller ^b^ (%)	1.1 ± 0.209	0.7–2.5	1.1 ± 0.237	0.4–2.4	0.965
Hind legs ^c^ (%)	34.1 ± 0.398	32.7–35.6	34.6 ± 0.317	33.5–36.1	0.390
Forelegs ^c^ (%)	26.4 ± 0.224	25.7–27.4	25.7 ± 0.443	23.4–27.1	0.189
Back ^c^ (%)	13.8 ± 0.486	11.1–15.2	15.2 ± 0.363	14.0–16.6	0.026
Ribs ^c^ (%)	14.4 ± 0.295	13.4–15.7	14.8 ± 0.324	13.8–16.5	0.312
Neck ^c^ (%)	11.3 ± 0.369	9.6–12.7	9.6 ± 0.284	8.4–10.7	0.005
Offal ^b^ (%)	35.7 ± 0.503	33.3–37.5	37.2 ± 1.042	33.7–42.5	0.227

^a^ Percentage of dead weight; ^b^ Percentage of dressed weight; ^c^ Percentage of cold carcass weight.

## Data Availability

Data can be found in Appendix A or from the corresponding author.

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
