# Peer review of "An Exploratory Study into the Influence of Sex on Body Measurements, Carcass Weights and Meat Yields of Giraffe (Giraffa camelopardalis angolensis)"

_foods, 2021, doi:10.3390/foods10102245_

Round 1

Reviewer 1 Report

This study evaluated the effect of sex on body position and carcass characteristics of giraffe. Since research on carcasses and meat qualities of game meat is limited regionally, it is thought that this manuscript presents meaningful and useful results. Moreover, experimental design and methodological approaches were performed appropriately according to the described purpose of this study. However, it is thought that further discussion is needed on the following two questions.

  1. In introduction, further explanation of the demand and perception of giraffe meat in the local area is needed. This is important for forming a consensus for recognizing giraffes as edible meat resources.
  2. L148 It is necessary to explain how the giraffe meat was confirmed in still pre-rigor status, since rigor status can considerably affect carcass weight and water-holding capacity (moisture loss in this study).

Editorial comment

  1. L200 “estimated lease squares means” it maybe “least” rather than “lease”.

Author Response

This study evaluated the effect of sex on body position and carcass characteristics of giraffe. Since research on carcasses and meat qualities of game meat is limited regionally, it is thought that this manuscript presents meaningful and useful results. Moreover, experimental design and methodological approaches were performed appropriately according to the described purpose of this study. However, it is thought that further discussion is needed on the following two questions.

  1. In introduction, further explanation of the demand and perception of giraffe meat in the local area is needed. This is important for forming a consensus for recognizing giraffes as edible meat resources.

This is a very valid point as the second Reviewer also indicated that the idea of eating giraffe meat was very disturbing, but the reality is that in developing countries, where game animals have value, they are looked after. Unfortunately, the mantra is “if the pay, they stay” which sometimes turns into “if the pay, we breed them” which is not always positive from a conservation viewpoint. However, the article is focused on the meat production potential of giraffe and not their conservation status. I have expanded on the way people view game meat in Africa, although this is all anecdotal as there is only one paper that I could find, from earlier work we did. Lines 51-58.

  1. L148 It is necessary to explain how the giraffe meat was confirmed in still pre-rigor status, since rigor status can considerably affect carcass weight and water-holding capacity (moisture loss in this study).

This is a valid point and we have explained this in the text (Lines 157-159).

Reviewer 2 Report

Report on the manuscript foods-1355933 entitled: An exploratory study into the influence of sex on body measurements, carcass weights and meat yields of giraffe (Giraffa Camelopardalis angolensis).

I must admit that, at the beginning, reading and thinking of giraffe carcass and meat was very disturbing.
From an objective point of view:

  • The age of the animals/carcasses should be included in the regression models as a continuous variable.
  • Table 2 (Person corr r) is ok (and also table 3) but further model regressions or model estimations should be proposed.
  • Same for Table 3: the age of the animals should be considered as a continuous variable too.

Author Response

  1. L200 “estimated lease squares means” it maybe “least” rather than “lease”.

Thank-you for highlighting this error.

Reviewer 2

Report on the manuscript foods-1355933 entitled: An exploratory study into the influence of sex on body measurements, carcass weights and meat yields of giraffe (Giraffa Camelopardalis angolensis).

I must admit that, at the beginning, reading and thinking of giraffe carcass and meat was very disturbing.

We respect this viewpoint and realise that the idea of eating giraffe would seem foreign to some people; I suppose it is like some people viewing the eating of horses? We appreciate that irrespective of this viewpoint, that you still conducted an objective review. The first reviewer also mentioned this perception and asked us to expand on how people perceive the eating of giraffe in the introduction, this has now been included in Lines 51-58.

From an objective point of view:

  • The age of the animals/carcasses should be included in the regression models as a continuous variable.
  • Table 2 (Person corr r) is ok (and also table 3) but further model regressions or model estimations should be proposed.
  • Same for Table 3: the age of the animals should be considered as a continuous variable too.

All three these comments have value if an accurate age of the animals were available. Unfortunately, the reality of culling wild animals is that you have to judge the age of an animal at a distance. The fact that one of the animals age, post mortem, was deemed to be nine years old, indicates this error in estimating age of the animals. We used Hall-Martin’s [3,13] data to calculate the age of the animals. When we fitted regression equations to the data as suggested, we had very low goodness of fits (R2 values – see the attached excel spreadsheet). We thus deemed it of little value to include these. None the less, we have elaborated on this shortcoming in the Conclusions (Line 448-450).

Round 2

Reviewer 2 Report

  • The introduction is too long. I think that the information about the long neck (from L. 59 to L.80, new version) is not needed.
  • 115-153 (new version). Please, include the abbreviations described in Figure 1 in the text.
  • It seems that the comment about the consideration of the age of the animals in the models was misunderstood.
    I understand the exclusion of the 9 years old animal from the dataset.

It is widely described in other species that the age of the animal affects its weight and growth. I think that the same happens in giraffes. The age of the animal should be considered as a continuous variable or as a covariable in the statistical models since there are not enough animals to consider “age” as another effect.
Considering the information of Appendix A, animals 6 and 7, which age was described as a range (4-6) showed the highest values for several variables… (proof of age effect)

Author Response

The introduction is too long. I think that the information about the long neck (from L. 59 to L.80, new version) is not needed.

We revisited this section and have decided to keep it in the text as our results showed no (statistically) difference in the neck measurements, indicating that our giraffe were still young and had not reached a maturity level where sexual dimorphism becomes measurable. See Lines 278-282.

115-153 (new version). Please, include the abbreviations described in Figure 1 in the text.

Thank-you for this suggestion, abbreviations have been included.

It seems that the comment about the consideration of the age of the animals in the models was misunderstood.
I understand the exclusion of the 9 years old animal from the dataset.

It is widely described in other species that the age of the animal affects its weight and growth. I think that the same happens in giraffes. The age of the animal should be considered as a continuous variable or as a covariable in the statistical models since there are not enough animals to consider “age” as another effect.
Considering the information of Appendix A, animals 6 and 7, which age was described as a range (4-6) showed the highest values for several variables… (proof of age effect).

We agree that like any other animal, age will have an effect on the body proportions, even more so after sexual maturity (the inflection point in the typical sigmoidal growth curve that animals typically exhibit).

After discussing this issue with the statistician Prof Martin Kidd (mkidd@sun.ac.za), he revisited the data, and it was then that we realised that we had described the statistical model incorrectly and that age had been included as covariate – this has now been corrected (Lines 216-218). We have also included Prof Kidd as a co-author. Thank-you for highlighting this omission (and your patience with this issue).